# Development of an Anisotropic Hyperelastic Material Model for Porcine Colorectal Tissues

**DOI:** 10.3390/bioengineering11010064

**Published:** 2024-01-08

**Authors:** Youssef Fahmy, Mohamed B. Trabia, Brian Ward, Lucas Gallup, Mary Froehlich

**Affiliations:** 1Department of Mechanical Engineering, Howard R. Hughes College of Engineering, University of Nevada, Las Vegas, NV 89154, USA; fahmy@unlv.nevada.edu (Y.F.); gallup@unlv.nevada.edu (L.G.); 2Department of Surgery, Kirk Kerkorian School of Medicine, University of Nevada, Las Vegas, NV 89154, USA; brian.ward@unlv.edu (B.W.); mary.froehlich@unlv.edu (M.F.)

**Keywords:** colorectal tissues, soft tissue modeling, experimental characterization

## Abstract

Many colonic surgeries include colorectal anastomoses whose leaks may be life-threatening, affecting thousands of patients annually. Various studies propose that mechanical interaction between the staples and neighboring tissues may play an important role in anastomotic leakage. Therefore, understanding the mechanical behavior of colorectal tissue is essential to characterizing the reasons for this type of failure. So far, experimental data characterizing the mechanical properties of colorectal tissue have been few and inconsistent, which has significantly limited understanding their behavior. This research proposes an approach to developing an anisotropic hyperelastic material model for colorectal tissues based on uniaxial testing of freshly harvested porcine specimens, which were collected from several age- and weight-matched pigs. The specimens were extracted from the same colon tract of each pig along their circumferential and longitudinal orientations. We propose a constitutive model combining Yeoh isotropic hyperelastic material with fibers oriented in two directions to account for the hyperelastic and anisotropic nature of colorectal tissues. Experimental data were used to accurately determine the model’s coefficients (circumferential, R^2^ = 0.9968; longitudinal, R^2^ = 0.9675). The results show that the proposed model can be incorporated into a finite element model that can simulate procedures such as colorectal anastomoses reliably.

## 1. Introduction

The gastrointestinal tract is one of the most intensively used organs in the human body. It ingests food and nutrients into the bloodstream while preventing toxic biproducts from damaging the body. Diseases such as inflammatory bowel disease, diverticulitis, and cancer may require colectomy to remove a section of the colorectal tract; over 320,000 patients undergo colorectal surgery in the United States annually. An anastomosis is then used to reconnect the colorectal tract and regain functionality. Recently, staples have been increasingly used instead of sutures, mainly due to the introduction of new and reliable disposable instruments and decreasing operative time [1]. Anastomotic leakage (AL) is one of the most common complications of colorectal anastomosis, increasing chances of morbidity [2]. Reports of AL have widely varied from 1.5 to 29.2%, e.g., in Refs. [2,3,4,5]. Hammond et al. [6] estimated that for every 1000 patients undergoing colorectal surgery, anastomotic leakage resulted in USD 28.6 million in additional costs and 9500 days of additional hospitalization. Patient physiology such as gender, obesity, nutrition, and inflammatory bowel disease predispose patients to AL. These factors interplay with technical factors such as stapled versus handsewn anastomoses or the use of side-to-side versus end-to-end connection [7].

While in vivo experimentation would best represent the conditions leading to AL, it may be difficult to conduct and control such experiments. Therefore, several researchers have attempted to simulate colon tissue and its interaction with the staples using finite element analysis [8,9,10,11,12]. This effort has been limited by a lack of reliable colorectal material models. Some of the efforts in this area are summarized below.

As Appendix A shows, colorectal tissues are complex with multiple layers. The tissues are hyperelastic with fiber bundles of varying strength that are distributed along different orientations. Due to the difficulties of harvesting human colons, relatively few researchers have been involved in this effort. Human colorectal tissues have been tested [13,14], and basic tensile properties of these tissues, along with comparisons of various harvesting locations within the colon tract have been presented. In a comparative study, researchers tested porcine colorectal tissue from 17 pigs and colon tissue harvested from 11 humans [15]. These specimens were frozen and thawed before the experiment. The results showed that the human and porcine tissues exhibited some level of similarity. However, human specimens had a lesser variability and a higher dependence on the harvested specimen’s location.

Despite the differences between the mechanical properties of human and porcine colorectal tissues, many researchers use porcine tissues as a substitute for human tissues because of both human and porcine colons being composed of four layers, the accessibility of porcine tissues, and their general physiological similarities to human tissues [16]. The following is a brief overview of the research in this area. Freshly harvested porcine rectal wall tissues were tested under uniaxial tension, compression, and shear [17]. The results indicated that the rectal wall is orthotropic. Ciarletta et al. [18] proposed a constitutive model for porcine colorectal tissue, combining an isotropic biological matrix and an anisotropic component, and describing the contributions of two longitudinal and circular muscular reinforcements. The results were compared with uniaxial and shear experiments of 20 specimens that were collected from five adult pigs. Uniaxial tensile testing of porcine colon specimens in three orientations from 15 large white pigs was conducted [19,20]. Harvested specimens were stored in a physiological saline solution at room temperature for six hours until testing convened. The parameters of a constitutive model for fiber-reinforced tissues were obtained. The authors of [21] used 15 cm segments of the porcine colons of ten pigs, which were stored in a 4 °C saline solution to conduct physiological inflation–extension tests. A constitutive model of the swine colon was validated against these experiments, concluding that a circumferential muscle layer does not provide significant mechanical support. Instead, the submucosal layer carries the circumferential load. Recently, the results of biaxial testing of different parts of the colonic tract harvested from several porcine specimens were reported [22]. Constitutive hyperelastic models that accounted for the multilayered conformation of the colonic wall and the fiber-reinforced configuration of the colorectal tissues were developed. A multiscale constitutive model to predict the anisotropic and nonlinear biomechanical behaviors of the multilayered colorectal tissues was developed based on experimental data [23]. Using biaxial tensile testing, the mechanical behavior of a porcine large intestine beyond ultimate stress was assessed [24]. The authors developed a multilayer anisotropic Holzapfel–Gasser–Ogden constitutive model, and a nonlinear damage constitutive model was applied to four fiber families. The model matched the experimental results well.

The aim of this work is to develop a porcine anisotropic hyperplastic model that is relatively simple and accurate such that it can be used in finite element modeling in the following steps: (a) conduct a series of uniaxial tests to further understand the stress–strain relationship of porcine colorectal tissues along the longitudinal and circumferential directions; (b) develop a material constitutive model that can present the experimental results; and (c) assess the effectiveness of these models for FEA simulations of the experiments.

## 2. Materials and Methods

### 2.1. Specimen Collection and Preparation

While experimentally testing each layer of colorectal tissues separately and developing a specific constitutive model for it may yield more insights, the process of separating the layers may add many uncertainties. Instead, freshly harvested porcine colorectal tissues were collected and were uniaxially tested using their whole thickness. Samples were cut along their circumferential and longitudinal directions.

The experiments were conducted on freshly harvested colon specimens from pigs of the F1 cross species (a Yorkshire and Landrace mix) from a local farm. This project was evaluated by the Institutional Animal Care and Use Committee at the University of Nevada, Las Vegas (UNLV), and deemed to be exempt since the policies of the institution apply to live animal care and animals sacrificed for research purposes. These colorectal tissues would have normally been discarded as in animals sacrificed for food consumption.

Harvested specimens came from five pigs between 3 and 8 months old, both male and female. The entire colons were collected immediately post mortem and brought to a UNLV laboratory where the experiments were conducted (Figure 1a). The procedure was started by draining the luminal contents by flushing the colon with tap water. The colon tract was then sliced longitudinally. Tissues were then flattened to cut each individual specimen for uniaxial testing (Figure 1b).

Specimens were cut along the longitudinal and circumferential directions to assess the colons’ anisotropic behavior. We developed a custom aluminum die cut with replaceable steel blades to ensure the consistency of the specimens’ rectangular shapes (Figure 2a). We used the die to cut specimens into 80 × 10 mm rectangles. Each colon tract yielded three to eight longitudinal and circumferential specimens, which were immediately placed in a saline solution until testing occurred. Experiments typically took place in less than four hours after harvesting.

To avoid slippage of the specimens during testing, they were secured to a sacrificial 3D printed serrated pair of clamps on each end using 3M VetBond^®^ (Saint Paul, MN, USA) surgical glue. Specimens and clamps were assembled in a custom jig with cylinders guiding the clamps’ placement (Figure 2b). A plateau supported the specimen free length to avoid sagging during specimen preparation. The nominal gage length and width of the specimens were 20 and 10 mm, respectively. The length, width, and thickness of each specimen were measured across the top, middle, and bottom before testing. Specimens were also weighed.

### 2.2. Testing Apparatus and Tensile Testing

The specimens were loaded on a custom-built low-force uniaxial tensile testing machine (Figure 3a). The apparatus used a stepper motor to apply a predetermined displacement time history. A 25N load cell (Interface^®^ SMT1-25N, Scottsdale, AZ, USA) measured the tensile force at a rate of 30 samples per second. A Back-Bone Modified GoPro Hero 10^®^ camera (San Mateo, CA, USA) with a Nikon AF-S Nikkor^®^ 18–140 mm Lens (Tokyo, Japan) was used to monitor the extension and reduction in the specimens’ width at a rate of 60 frames per second. Videos were recorded at 5.3 k resolution, which translates to 15.8 MP for each frame. A black curtain covered the area behind the specimen to reduce optical noise. Two universal joints connected the clamps to the machine to reduce the possibility of misalignment (Figure 3b).

Table 1 lists the number of samples used for each test and the average unloaded initial dimensions. Testing began with loading each specimen through four pre-conditioning cycles to a deformation corresponding to 7% strain at a rate of 0.05 mm/s. Once these cycles were completed, the specimen was loaded uniformly using the same displacement rate until it failed (Figure 4a,b). Figure 4c shows a typical load time history of a colorectal specimen.

**Table 1 bioengineering-11-00064-t001:** Pre-conditioning dimensions of the porcine colorectal specimens.

Tissue Orientation	Circumferential Direction	Longitudinal Direction
Number of Specimens Tested	20	18
Gauge Length ^1^, mm	20	20
Average Width (Std. Dev.), mm	10.1 (1.7)	8. 6 (1.6)
Average Thickness (Std. Dev.), mm	1.0 (0.2)	1.1 (0.3)
Average Weight (Std. Dev.), g	1.2 (0.2)	1.2 (0.4)

^1^ No variation because a die was used to cut all specimens.

### 2.3. Data Processing

The data were filtered using a 4th-order polynomial to smooth the effect of the stepper motor signals. A 5th-order lowpass Butterworth filter with a cutoff frequency of 800 Hz smoothed the slope of the load data, which was monitored and used to identify the initiation of the post-conditioning phase. Time and load were zeroed at this point.

Each image was converted to grayscale and then to black and white. The pixels that correspond to the boundaries of the tissue were identified and separated into four groups, which described the four edges of the specimen. Figure 5 shows this process. A custom Matlab^®^ (Natick, MA, USA) code monitored the changes in the specimens’ extension and width during the experiment. The minimum width at the first instant of the post-conditioning phase was used for stress calculation.

### 2.4. Proposed Constitutive Model

As shown in Section 1, the experimental results generally indicated that porcine colorectal tissues exhibit an anisotropic, nonlinear mechanical behavior during large deformation. This work aims to use experimental data of porcine colorectal tissues to develop a constitutive model that can accurately relate the stresses and strains within the colorectal tissues, which can predict the colorectal tissue behavior under various loadings.

In addition to the review of the colorectal constitutive models presented in Section 1, a comprehensive review of these models can be found in various papers including by Refs. [25,26]. The proposed model assumes that colorectal tissues are isotropic and hyperelastic with fibers embedded uniformly in the submucosa and the muscularis propria. The first step in developing the constitutive model started by proposing an expression for the strain energy, which controls the relation between load and deformation. Multiple researchers have suggested several forms for biological tissues. This research proposes an anisotropic hyperelastic polynomial fiber-reinforced composite model. This concept is based on the observation of Ref. [27] of fiber bundles within mice intestines. The analysis presented in this section is based on Refs. [28,29]. Related analysis can be also found in [10].

If the deformed state in terms of a coordinate system X is defined as *x*, then the after-deformation gradient of the body is F=∂x/∂X. The volume ratio, *J*, is *J*$=\left|F\right|.\;$The Cauchy strain tensor is defined as the following:(1)$$C=F^{T}F$$

The deformation gradient can be split into volumetric and distortional deformation such that, $F={\left(J\right)}^{1/3}\bar{F}$. Substituting into Equation (1), the Cauchy strain tensor becomes
(2)$$C={\left(J\right)}^{2/3}\bar{C}$$
where $\bar{C}=\left[\begin{array}{ccc} {\lambda_{1}}^{2} & 0 & 0\\ 0 & {\lambda_{2}}^{2} & 0\\ 0 & 0 & \lambda_{3}^{2} \end{array}\right]$, $\lambda_{i}$ is the stretch in the i-th principal direction.

The strain energy function can be expressed as a combination:(3)$$\Psi =\Psi_{iso}+\Psi_{aniso}$$
where $\Psi_{iso}$ and $\Psi_{aniso}$ are the isotropic and the anisotropic strain energies, respectively. These terms can be expressed as the following:(4)$$\Psi_{iso}=\sum_{i=1}^{n}a_{i}{\left({\bar{I}}_{1}-3\right)}^{i}$$
(5)$$\Psi_{aniso}=\sum_{j=2}^{m}c_{j}{\left({\bar{I}}_{4}-1\right)}^{j}+\sum_{k=2}^{m}e_{k}{\left({\bar{I}}_{6}-1\right)}^{k}$$

$a_{i}$ are parameters that specify the material constants while $c_{j}$ and $e_{k}$ are the fiber stiffness parameters, respectively. The strain energy functions are expressed in terms of the first strain invariants and two pseudo-invariants as the following:(6)$${\bar{I}}_{1}=tr\left(\bar{C}\right)$$
(7)$${\bar{I}}_{4}=a\cdot \bar{C}a$$
(8)$${\bar{I}}_{6}=g\cdot \bar{C}g$$

The fiber directions are defined using these two vectors:$$a=\left[\begin{array}{ccc} cos\left(\beta_{1}\right) & sin\left(\beta_{1}\right) & 0 \end{array}\right]$$
$$g=\left[\begin{array}{ccc} cos\left(\beta_{2}\right) & sin\left(\beta_{2}\right) & 0 \end{array}\right]$$
where $\beta_{1}$ and $\beta_{2}$ are the undeformed angles of the two fiber sets in terms of the coordinate system X. The isotropic energy function is based on the constitutive model Yeoh proposed for hyperelastic materials [30]. $c_{j},$ $e_{k}$, $\beta_{1}$, and $\beta_{2}$ describe the colorectal tissues’ anisotropic behavior.

To determine the constitutive equations, the strain energy function, Ψ, was differentiated with respect to $\bar{C}$. The second Piola–Kirchhoff stress, S, can be determined as
(9)$${\;S=J}^{-2/3}\left(\tilde{S}-\frac{1}{3}\left(\bar{C}:\tilde{S}\right){\bar{C}}^{-1}\right)$$
(10)$$\tilde{S}=2\left(\sum_{i=1}^{n}\frac{\partial \Psi_{iso}}{\partial {\bar{I}}_{1}}\frac{\partial {\bar{I}}_{1}}{\partial \bar{C}}+\sum_{j=2}^{m}\frac{\partial \Psi_{iso}}{\partial {\bar{I}}_{4}}\frac{\partial {\bar{I}}_{4}}{\partial \bar{C}}+\sum_{k=2}^{n}\frac{\partial \Psi_{iso}}{\partial {\bar{I}}_{6}}\frac{\partial {\bar{I}}_{6}}{\partial \bar{C}}\right)$$

In the case of uniaxial tension with no compressibility, *J*=1, which means that $\lambda_{1}=\lambda$ and $\lambda_{2}=\lambda_{3}=\lambda^{-1/2}$. This loading can be described by adding p, which is a pressure determined by satisfying the boundary conditions, so Equation (10) becomes
(11)$$S=p\left[\begin{array}{ccc} \lambda^{-2} & 0 & 0\\ 0 & \lambda & 0\\ 0 & 0 & \lambda \end{array}\right]+\left[\begin{array}{ccc} {\tilde{S}}_{11} & {\tilde{S}}_{12} & {\tilde{S}}_{13}\\ {\tilde{S}}_{21} & {\tilde{S}}_{22} & {\tilde{S}}_{23}\\ {\tilde{S}}_{31} & {\tilde{S}}_{32} & {\tilde{S}}_{33} \end{array}\right]-\frac{\lambda^{2}{\tilde{S}}_{11}+\lambda^{-1}\left({\tilde{S}}_{22}+{\tilde{S}}_{33}\right)}{3}\left[\begin{array}{ccc} \lambda^{-2} & 0 & 0\\ 0 & \lambda & 0\\ 0 & 0 & \lambda \end{array}\right]$$
p can be determined by considering the equation corresponding to $S_{33}$, which are equal to zero. p can be then substituted in the $S_{11}$ equation to obtain the following constitutive equation:(12)$$S_{11}={\tilde{S}}_{11}-\frac{{\tilde{S}}_{33}}{3\lambda^{3}}$$
where
$${\tilde{S}}_{11}=2\left(\sum_{i=1}^{n}a_{i}{i\left({\bar{I}}_{1}-3\right)}^{i-1}+\left(\sum_{j=2}^{m}c_{j}j{\left({\bar{I}}_{4}-1\right)}^{j-1}\right){cos}^{2}\left(\beta_{1}\right)+\left(\sum_{k=2}^{m}e_{k}{k\left({\bar{I}}_{6}-1\right)}^{k-1}\right){cos}^{2}\left(\beta_{2}\right)\right)$$
$${\tilde{S}}_{33}=2\sum_{i=1}^{n}a_{i}{i\left({\bar{I}}_{1}-3\right)}^{i-1}$$

The true stress can be determined based on the second Piola–Kirchhoff stress, S [31], as
(13)$$\sigma_{i}={\;J}^{-1}F\left(S\right)F^{T}$$

Since testing was conducted along the circumferential and longitudinal directions, it was relatively easy to repeat the above process for the longitudinal data. Assuming that these directions are 1 and 2, respectively, the stress in the case of longitudinal specimens can be expressed using this equation:$${\tilde{S}}_{22}=2\left(\sum_{i=1}^{n}a_{i}{i\left({\bar{I}}_{1}-3\right)}^{i-1}+\left(\sum_{j=2}^{m}c_{j}j{\left({\bar{I}}_{4}-1\right)}^{j-1}\right){sin}^{2}\left(\beta_{1}\right)+\left(\sum_{k=2}^{m}e_{k}{k\left({\bar{I}}_{6}-1\right)}^{k-1}\right){sin}^{2}\left(\beta_{2}\right)\right)$$

The second Piola–Kirchhoff stress in this case is
(14)$$S_{22}={\tilde{S}}_{22}-\frac{{\tilde{S}}_{33}}{3\lambda^{3}}$$

With these equations, the problem became that of determining the set of the constitutive model variables ($a_{i}$, $c_{j}$, $e_{k}$, $\beta_{1},$ and $\beta_{2}$) that gave the best fit to the experimental results.

## 3. Results

### 3.1. Experimental Results

Conditioning affected the dimensions of the specimens that were already measured after harvesting (Table 1). As Table 2 shows, specimens became longer while their widths were reduced. These post-conditioning dimensions were used in the remainder of this work.

Specimens were loaded during the post-conditioning phase of the experiment until approximately 50% of engineering strain was achieved. However, it was observed that specimens experienced inconsistent tearing of their tissues and fibers before reaching the target strain. We therefore decided to limit the reported results to approximately 20% of engineering strain since tissue damage typically occurred after this value. As mentioned earlier, engineering strain and stress were calculated based on the post-conditioning length and the minimum width of each specimen at the onset of the post-conditioning phase. The minimum width was obtained from the image processing presented in Section 2.4. Figure 6 shows the individual specimens’ engineering strain and stress in the circumferential and longitudinal directions. Additionally, Figure 7 shows the average and standard deviation engineering stress for both directions.

### 3.2. Constitutive Model Results

The experimental results of Section 3.1 were combined with the anisotropic hyperelastic model of Section 2.4 to determine the eleven coefficients ($a_{i}$, $c_{j}$, $e_{k}$, $\beta_{1},$ and $\beta_{2}$) that could best describe the mechanical behavior of the porcine colorectal tissues. The problem had two objective functions that corresponded to the normalized errors of the constitutive model with respect to the experimental circumferential and longitudinal stress–strain curves:(15)$$E_{c}=\frac{\sum_{l=1}^{L}{\left({\sigma_{ce}}_{l}-{\sigma_{cm}}_{l}\right)}^{2}}{\sum_{l=1}^{L}{\left({\sigma_{ce}}_{l}\right)}^{2}}$$
(16)$$E_{t}=\frac{\sum_{q=1}^{Q}{\left({\sigma_{te}}_{l}-{\sigma_{tm}}_{l}\right)}^{2}}{\sum_{q=1}^{Q}{\left({\sigma_{te}}_{l}\right)}^{2}}$$
where $\sigma_{ce}$ and $\sigma_{te}$ are the experimental circumferential and longitudinal stresses, respectively. *L* and *Q* are the number of circumferential and longitudinal stress–strain datasets, respectively.

Using the Genetic Algorithm (GA) Toolbox within Matlab^®^ (R2022a) a Pareto front for the two objective functions was generated. As expected, the extremes of the Pareto front corresponded to negligible errors in one direction and large errors in the other. A cluster of points in the lower left corner of the Pareto front corresponded to acceptable solutions (Figure 8). Figure 9 shows various results of the Pareto front while Figure 9a,b show solutions in which a good stress–strain representation of either the circumferential or longitudinal directions led to large errors in the other direction. Figure 9c is a case with the most balanced circumferential and longitudinal errors. Table 3 lists the coefficients corresponding to this case. Statistical analysis (circumferential, R^2^ = 0.9968; longitudinal, R^2^ = 0.9675) indicated that the model’s coefficients represented the experimental data accurately by accounting for the hyperelasticity of the tissues while accounting for the different strengthening effects of the fibers oriented in multiple directions.

### 3.3. Finite Element Verification

To assess the effectiveness of the proposed approach, we included the data of Table 3 which were in an ANSYS^®^ Parametric Design Language (APDL) (Canonsburg, PA, USA), which was incorporated in two finite element models (FEMs) with volumes that corresponded to the average sizes of the post-conditioning samples in the circumferential and longitudinal directions (Table 1 and Table 2). Both models, which were simulated in ANSYS^®^ Mechanical 2021 R2, used 0.25 mm hexadecimal cubic elements with the large deformation condition turned on (Figure 10). This mesh size was verified to be numerically stable when compared to meshes using 0.5 mm and 0.125 mm. Local coordinate frames described the fiber orientation effects in both cases. The top end of Figure 10 was completely fixed while the opposite one was subjected to a prescribed deflection of 6 mm. Stresses normal to the loading direction were recorded by the model, as well as the reaction force at the fixed end. Figure 11 shows the ANSYS stress outputs at the end of the simulation.

Figure 12a,b compare the true stress and reaction forces of the experiment, constitutive model, and finite element in the circumferential and longitudinal directions, respectively. The reported FEM stresses correspond to the average for elements at the midsection of the specimen while the force was recorded at the fixed support. Table 4 lists the normalized errors of the proposed constitutive model with respect to the experimental circumferential and longitudinal stress–strain curves.

## 4. Discussion

During the post-conditioning phase of the experiment, various signs of tissue and fiber damage were evident, typically near the middle of specimens. Tearing was preceded by thickness reduction as well as a change in the color and opaqueness of the specimens in many cases. As Figure 6 and Figure 7 show, the tearing mechanisms were different in the circumferential and longitudinal specimens, with longitudinal specimens being more consistent and less affected by tearing. This observation is further strengthened by comparing the standard deviation in both cases. Additionally, these two figures showed that colorectal tissues are significantly stronger in the circumferential direction than in the longitudinal direction, which is consistent with the orientation of the colorectal tissues’ main fibers. Additionally, while circumferential stress–strain curve was almost linear, the longitudinal stress–strain exhibited more nonlinearity.

Table 5 summarizes some of the results of other researchers and compares them to ours. The table clearly shows a large variation in the stress–stain curves in the circumferential and longitudinal directions, qualitatively and quantitatively. This variation may be explainable by the fact that some researchers continued testing to relatively high strains or until complete failure [15,17,18]. As observed above, tearing the fibers may make the results difficult to assess. Alternatively, some researchers opted to test specimens until they achieved relatively small strain values [22,23]. The closest tested strain range was by Ref. [20]. Five specimens were tested in each orientation in this case. However, the results were softer than what we reported. The varying results reported here may be secondary to the processing of specimens prior to testing. We believe that the freshness and lack of processing of our specimens makes our research more relevant.

Table 3 indicates that the GA and Pareto results suggest that the fibers were almost normal to each other and aligned with the circumferential and longitudinal directions. Comparing $a_{1}$, $c_{2}$, and $e_{2}$ showed that the circumferential fibers dominated the corresponding coefficients of the Yeoh hyperplastic model and the longitudinal fibers. Similarly, the coefficients of the higher-order terms, $a_{3}$, $c_{4}$, and $e_{4}$, were significantly smaller. However, removing them from the model reduced its accuracy.

Figure 12 shows that the FEM results matched those of the experiments and the constitutive model well. The variation in the FEM force results in the circumferential direction may be attributed to the nonuniformity of stresses, especially at the corners. From a clinical perspective, this validates the common surgical practice of reinforcing the ends of a stapled anastomosis with an additional stitch. Similarly, given the orientation of the stapler, anastomoses are often performed in a side-to-side orientation. This allows for a large common channel that stretches in the circumferential direction instead of longitudinally. Prior to the advent of staplers, handsewn anastomoses were performed in an end-to-end orientation, which relied on longitudinal tissue strength.

The current study was subject to several limitations. First, the study could benefit from testing a larger number of porcine colorectal tissues harvested from different regions. Additionally, conducting the uniaxial testing at various orientations besides circumferential and longitudinal will provide additional insights regarding the fibers’ role. To accurately describe the complex structure of colorectal tissues, a layer-specific constitutive model may be useful. In such a model, each layer could have its own hyperelastic characteristics and fibers with their own specific orientations and strength. However, such a model may be computationally intensive and difficult to incorporate in a finite element model. Furthermore, the en bloc behavior of the tissues reported in this work may be much more clinically relevant. Finally, the effects of the residual stresses in porcine tissues were not considered. Understanding the role of residual stresses may help to better understand the behavior of in vivo colorectal tissues.

## 5. Conclusions

The behavior of colorectal tissues under loading is important to understand, especially the reasons for why colorectal anastomoses can fail. In this work, uniaxial testing in the circumferential and longitudinal directions of freshly harvested porcine colorectal tissues was conducted. Engineering stress–strain relationships in both directions were obtained based on a combination of data analyses and image processing. This work also presents a hyperelastic anisotropic constitutive model of colorectal tissues. This model combined the Yeoh hyperplastic material model with two sets of anisotropic fibers that were oriented within the tissue. Three-term polynomials represented the hyperelastic behavior and the fibers in terms of the strain invariants and two pseudo-invariants. A multi-objective optimization approach identified the parameters of these three models as well as the fiber angles. The constitutive model was incorporated in finite element models to simulate experiments. The constitutive model and FEMs matched the experimental results well, which indicates that the proposed constitutive models can be used to further understand the mechanics of the colon and to better guide optimal surgical technique. This proposed constitutive model may help explain the clinical findings of an increased leak rate with handsewn anastomosis compared to stapled. It can also help surgeons construct colorectal anastomoses that are more resistant to leakage and aid in the design of stapler devices that decrease anastomotic leak rates.

## Figures and Tables

**Figure 1 bioengineering-11-00064-f001:**
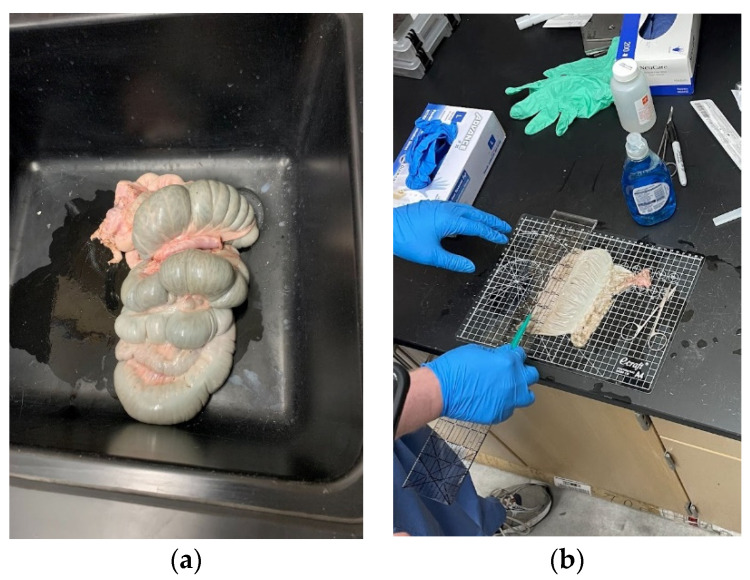
Specimen preparation: (**a**) colon extraction and (**b**) colon tract slicing and flattening.

**Figure 2 bioengineering-11-00064-f002:**
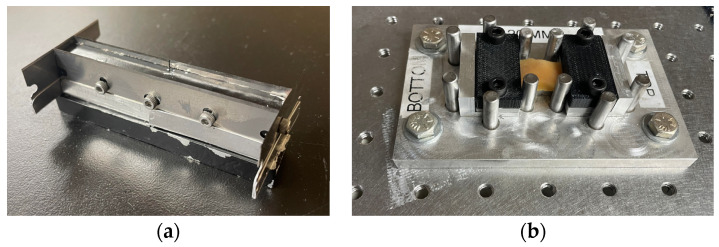
Specimen preparation. (**a**) Die cut to prepare specimens. (**b**) Colorectal tissue specimen glued to two sacrificial clamps. A jig was used to ensure consistent specimens. Table 1 includes the specimens’ dimensions.

**Figure 3 bioengineering-11-00064-f003:**
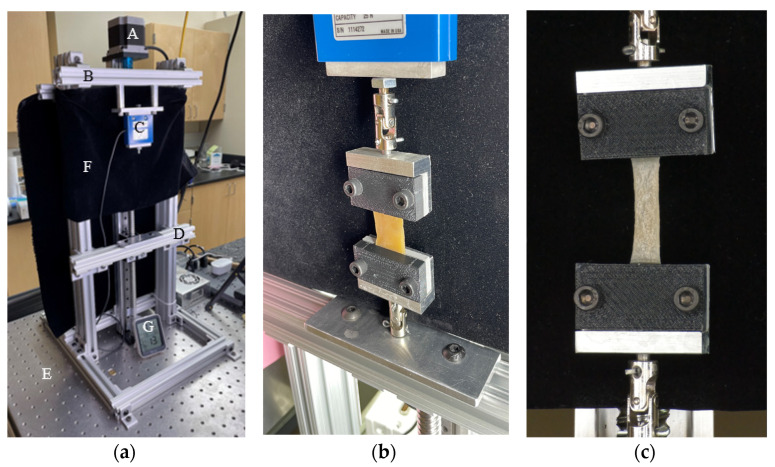
Uniaxial testing machine. (**a**) Components: A. stepper motor; B. fixed frame; C. load cell; D. moving frame; E. optical table; F. black curtain; G. environmental monitor. (**b**) A specimen under loading with universal joints connected to the clamps. (**c**) View from the camera. Table 1 includes the specimens’ dimensions.

**Figure 4 bioengineering-11-00064-f004:**
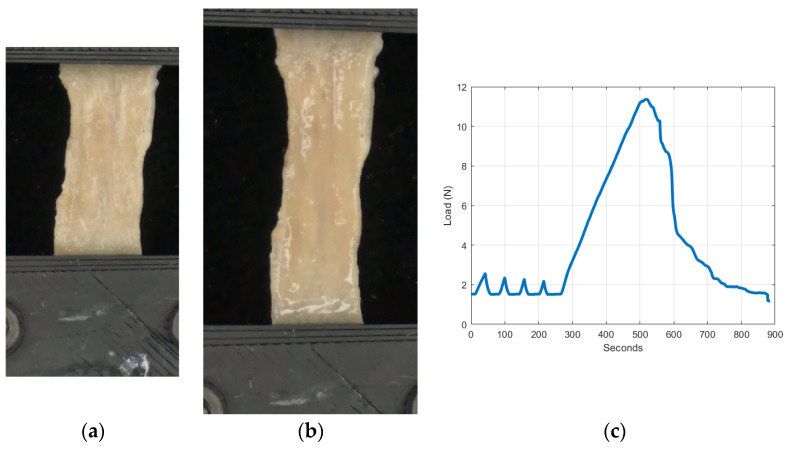
Uniaxial testing of colorectal tissues. (**a**) Initial instant of a specimen in the post-conditioning phase. (**b**) Final instant of a specimen in the post-conditioning phase immediately before the tissue tearing. (**c**) A typical colorectal tissue load time history.

**Figure 5 bioengineering-11-00064-f005:**
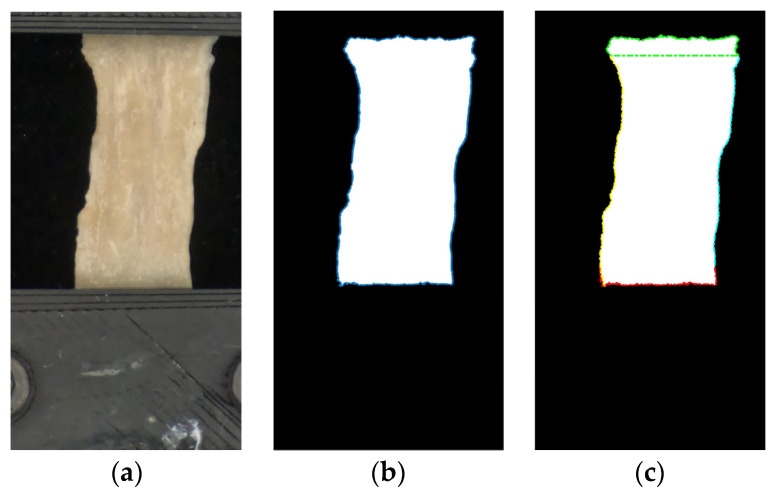
Monitoring the changes in elongation and width of the specimens during uniaxial testing. (**a**) Original image. (**b**) Black and white equivalent image; boundaries were identified. (**c**) Black and white equivalent image with the four boundaries separated into four regions.

**Figure 6 bioengineering-11-00064-f006:**
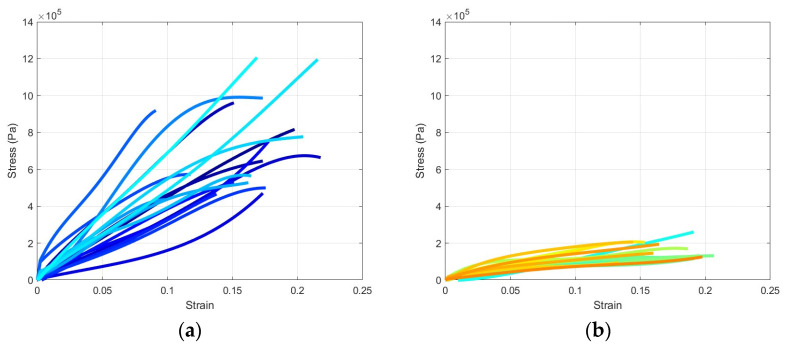
Engineering stress–strain curves: (**a**) circumferential direction and (**b**) longitudinal direction.

**Figure 7 bioengineering-11-00064-f007:**
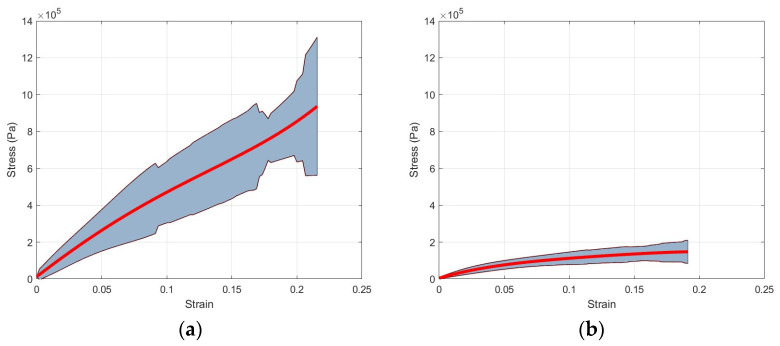
Average and standard deviations of the engineering stress–strain curves: (**a**) circumferential direction and (**b**) longitudinal direction.

**Figure 8 bioengineering-11-00064-f008:**
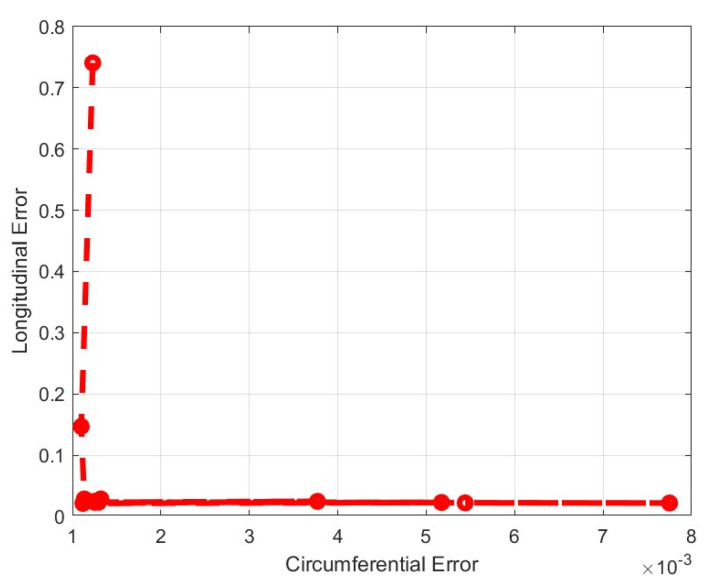
Pareto front for the multi-objective minimization of the anisotropic hyperelastic constitutive porcine colorectal tissue model.

**Figure 9 bioengineering-11-00064-f009:**
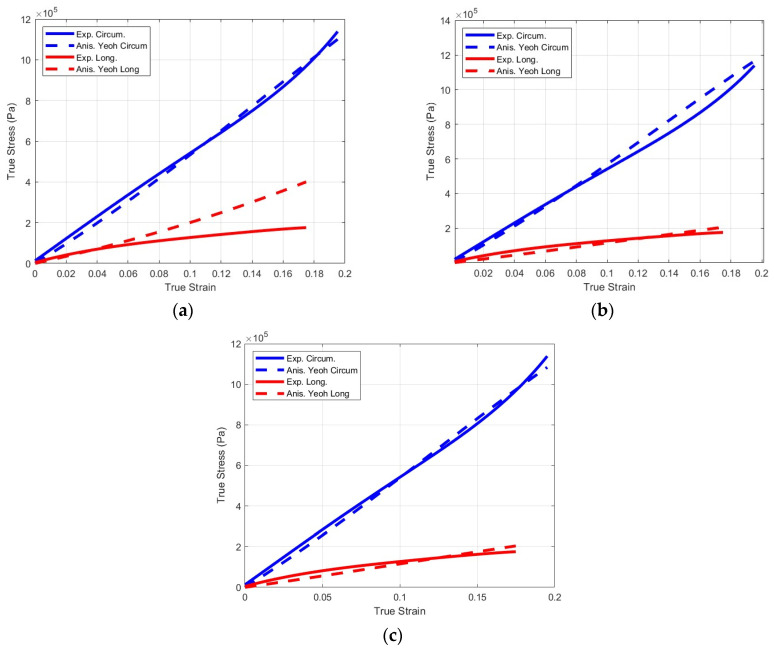
Various examples of the multi-objective minimization of the anisotropic hyperelastic constitutive porcine colorectal tissue model. (**a**) No consideration of the longitudinal stress–strain data. (**b**) No consideration of the circumferential stress–strain data. (**c**) An example of an acceptable solution.

**Figure 10 bioengineering-11-00064-f010:**
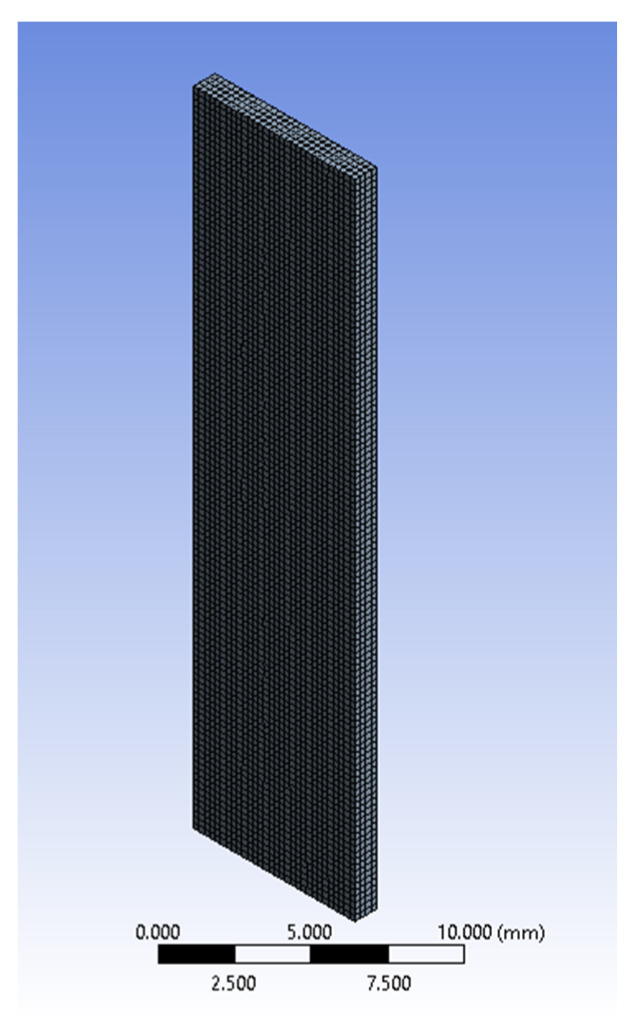
A meshed FEM model of a post-conditioning specimen.

**Figure 11 bioengineering-11-00064-f011:**
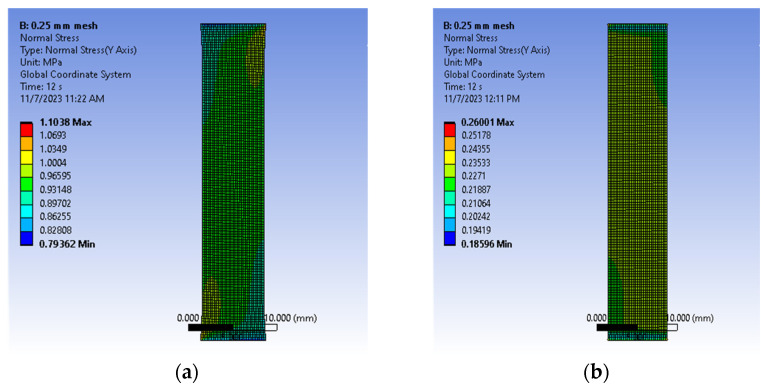
Comparison of stress normal to the direction of loading at the end of the FEM simulations: (**a**) circumferential and (**b**) longitudinal.

**Figure 12 bioengineering-11-00064-f012:**
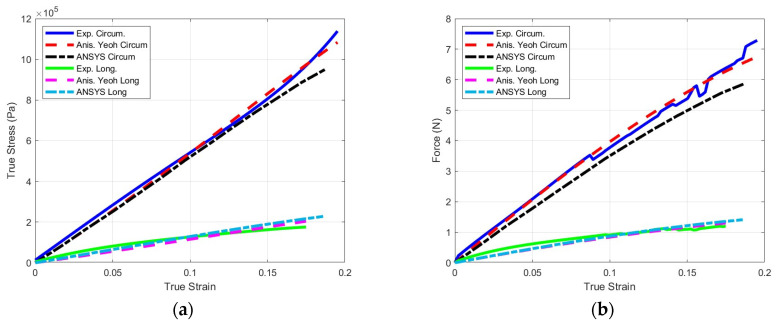
Comparison of the circumferential and longitudinal results of the experiments, proposed model, and finite element analysis: (**a**) true stress and strain and (**b**) reaction force.

**Table 2 bioengineering-11-00064-t002:** Post-conditioning dimensions of the porcine colorectal specimens.

Tissue Orientation	Circumferential Direction	Longitudinal Direction
Number of Specimens Tested	20	18
Length (Std. Dev.), mm	29.0 (2.6)	29.8 (2.2)
Average Width (Std. Dev.), mm	8.9 (2.0)	8.0 (2.1)

**Table 3 bioengineering-11-00064-t003:** Coefficients of the anisotropic hyperelastic constitutive porcine colorectal tissue model of Figure 9c.

$a_{1}$ **(Pa)**	$a_{2}$ **(Pa)**	$a_{3}$ **(Pa)**	
1.400 × 10^5^	−9.585 × 10^3^	−1.976 × 10^3^	
$c_{2}$ **(Pa)**	$c_{3}$ **(Pa)**	$c_{4}$ **(Pa)**	$\beta_{1}$ **(Degrees)**
4.929 × 10^5^	−2.981 × 10^5^	6.553 × 10^3^	0.790
$e_{2}$ **(Pa)**	$e_{3}$ **(Pa)**	$e_{4}$ **(Pa)**	$\beta_{2}$ **(Degrees)**
3.0146 × 10^4^	−2.9556 × 10^4^	−739.324	90.100

**Table 4 bioengineering-11-00064-t004:** Comparison of the normalized errors of the proposed constitutive model and FEM results with the experimental circumferential and longitudinal stress–strain.

	$E_{c}$	$E_{t}$
Proposed Constitutive Model	0.0011	0.0212
FEM	0.0048	0.0061

**Table 5 bioengineering-11-00064-t005:** Comparison of stress–strain curves porcine colorectal tissues available in the literature with the current work’s results.

Source		Qualitative Description	Maximum Reported Engineering Strain and Stress (Pa)	Engineering Stress around 0.2 Strain (Pa)
[15]	Circumferential	Nonlinear until 0.05 strain, followed by a linear rise	1.06, 5.1 × 10^5^	–
	Longitudinal		1.20, 6.3 × 10^5^	–
[17]	Circumferential	Nonlinear until 0.25 strain, followed by a linear rise	0.90, 1.2 × 10^5^	1.0 × 10^4^
	Longitudinal	Nonlinear	0.90, 3.3 × 10^5^	3.0 × 10^3^
[18]	Circumferential	Nonlinear	0.84, 3.2 × 10^5^	2.5 × 10^4^
	Longitudinal	Linear until about 0.08 strain, followed by a linear rise	0.23, 8.8 × 10^3^	8.5 × 10^3^
[20]	Circumferential	Nonlinear	0.40, 3.20 × 10^5^	1.0 × 10^5^
	Longitudinal	Linear	0.40, 7.5 × 10^4^	3.5 × 10^4^
[22]	Circumferential	Nonlinear	0.08, 6.0 × 10^5^	–
	Longitudinal	Nonlinear	0.03, 4.0 × 10^5^	–
[23]	Circumferential	Nonlinear	0.11, 9.1 × 10^4^	–
	Longitudinal	Nonlinear	0.04, 9.6 × 10^4^	–
Current Study	Circumferential	Linear	0.22, 9.0 × 10^5^	8.2 × 10^5^
	Longitudinal	Nonlinear	0.18, 1.7 × 10^5^	–

## Data Availability

The data presented in this study are available upon request from the corresponding author.

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
