# Peer review of "Development of an Anisotropic Hyperelastic Material Model for Porcine Colorectal Tissues"

_bioengineering, 2024, doi:10.3390/bioengineering11010064_

Round 1
Reviewer 1 Report
Comments and Suggestions for Authors
Dear Author
1-The is very well written; clear, precise, and easy to understand. But it has lack of Institutional Review Board Statement and ethical naming the Committee and number.
2- The manuscript has novelthy consideration. What is the advantage of the research to the previous publication:
Puértolas S, Peña E, Herrera A, Ibarz E, Gracia L. A comparative study of hyperelastic constitutive models for colonic tissue fitted to multiaxial experimental testing. Journal of the mechanical behavior of biomedical materials. 2020 Feb 1;102:103507.
Bhattarai A, May CA, Staat M, Kowalczyk W, Tran TN. Layer-specific damage modeling of porcine large intestine under biaxial tension. Bioengineering. 2022 Oct 6;9(10):528.
Ciarletta P, Dario P, Tendick F, Micera S. Hyperelastic model of anisotropic fiber reinforcements within intestinal walls for applications in medical robotics. The International Journal of Robotics Research. 2009 Oct;28(10):1279-88.
Comments on the Quality of English Language
Minor editing of English language required
Reviewer 2 Report
Comments and Suggestions for Authors
In this study, the authors have introduced a relatively accurate porcine anisotropic hyperplastic model to simulate colorectal anastomoses. The work is valuable. However, unfortunately, it can’t address the scope of the journal?
But, authors can address the issues for second submission to another journal:
- Authors should enrich the abstract with more quantitative results.
- The text is written in passive form. Please revise the text and correct some grammatical mistake.
- In Line 4, rearrange the order of reference numbers.
- Revise lines 89-90 for citing references.
- 2.1. “Colon Tissue Structure” is not a method and should be summarized.
- In discussion section, authors should compare the results with similar works under a literature survey.
- Author should mention the limitation of the previous works, challenges and opportunities.
- Authors must address the ethical concerns about this study.
Best
Comments on the Quality of English LanguageExtensive editing of English language required
Reviewer 3 Report
Comments and Suggestions for Authors
Good work.
Please write abstract with proper result indication (statistics).
Add animal protocol approval number at proper place.
Why pig tissue selected while rodent tissues and cancer research studies conducted on rodent?
Bibliography is not proper, do as per the journal guidelines.
Reviewer 4 Report
Comments and Suggestions for Authors
This paper generated models for the colon based on cadaveric porcine tissues post-mechanical stress testing. There is a lack of quantitative comparisons throughout the study, which limits the rigor and reproducibility of the findings.
-Please revise the Methods to describe the materials and methods. Fig.1 showing the structure of the human intestine is not appropriate for a primary research article and would be better in a review article. Likewise, the photographs included in Fig. 2 are not suitable for publication. Please edit.
-The descriptions and figures of the experimental setup (Fig. 2-4) would benefit from inclusion of dimensions and scaling.
-Throughout the results, please describe the findings in terms of quantitative comparisons with proper statistics included (mean +/- stdev, p-value, n).
Comments on the Quality of English LanguageMinor edits needed. The in-text citations should be reformatted per journal requirements and the comma before the number should be removed and multiple citations should be combined (example [4-8]).
Round 2
Reviewer 2 Report
Comments and Suggestions for Authors.
Comments on the Quality of English LanguageMinor checking is needed.